# Regional Strain Pattern Index—A Novel Technique to Predict CRT Response

**DOI:** 10.3390/ijerph18030926

**Published:** 2021-01-21

**Authors:** Michał Orszulak, Artur Filipecki, Wojciech Wróbel, Adrianna Berger-Kucza, Witold Orszulak, Dagmara Urbańczyk-Swić, Wojciech Kwaśniewski, Edyta Płońska-Gościniak, Katarzyna Mizia-Stec

**Affiliations:** 1First Department of Cardiology, School of Medicine in Katowice, Medical University of Silesia, ul. Ziolowa 45/47, 40-635 Katowice, Poland; arturfilipecki@wp.pl (A.F.); wojtekwrobel@poczta.onet.pl (W.W.); adaberger@wp.pl (A.B.-K.); worszulak@poczta.onet.pl (W.O.); dagau@poczta.onet.pl (D.U.-S.); wkwasniewski@op.pl (W.K.); kmiziastec@gmail.com (K.M.-S.); 2Department of Cardiology, Pomeranian Medical University, 70-204 Szczecin, Poland; edytaplonska@life.pl

**Keywords:** cardiac resynchronization therapy, dyssynchrony, strain pattern, heart failure, RSPI

## Abstract

Background: Cardiac resynchronization therapy (CRT) improves outcome in patients with heart failure (HF) however approximately 30% of patients still remain non-responsive. We propose a novel index—Regional Strain Pattern Index (RSPI)—to prospectively evaluate response to CRT. Methods: Echocardiography was performed in 49 patients with HF (66.5 ± 10 years, LVEF 24.9 ± 6.4%, QRS width 173.1 ± 19.1 ms) two times: before CRT implantation and 15 ± 7 months after. At baseline, dyssynchrony was assessed including RSPI and strain pattern. RSPI was calculated from all three apical views across 12 segments as the sum of dyssynchronous components. From every apical view, presence of four components were assessed: (1) contraction of the early-activated wall; (2) prestretching of the late activated wall; (3) contraction of the early-activated wall in the first 70% of the systolic ejection phase; (4) peak contraction of the late-activated wall after aortic valve closure. Each component scored 1 point, thus the maximum was 12 points. Results: Responders reached higher mean RSPI values than non-responders (5.86 ± 2.9 vs. 4.08 ± 2.4; *p* = 0.044). In logistic regression analysis value of RSPI ≥ 7 points was a predictor of favorable CRT effect (OR: 12; 95% CI = 1.33–108.17; *p* = 0.004). Conclusions: RSPI could be a valuable predictor of positive outcome in HF patients treated with CRT.

## 1. Introduction

Cardiac resynchronization therapy (CRT) improves the outcome and reduces symptoms in advanced Heart Failure with a Reduced Ejection Fraction (HFrEF) patients with a prolonged duration of QRS complex, especially with Left Bundle Branch Block (LBBB) [1,2]. Pooled data from six selected studies showed that CRT reduced all-cause mortality by 28% and new hospitalizations for worsening HF by 37% [3]. In the CARE-HF and COMPANION trials, it has been shown that CRT reduces heart failure symptoms by one NYHA class and improves quality of life by 5–6% [1,4]. The main purpose of CRT is to reduce dyssynchrony and to restore physiological activation of the left ventricle (LV) myocardium [5]. A thorough, reliable evaluation of dyssynchrony seems to be crucial in the selection of CRT recipients [6]. The precise measurement of intraventricular dyssynchrony must take into account its two main components—electrical dyssynchrony and mechanical dyssynchrony, which are complementary because the final effect of resynchronization therapy depends on the presence of both [7]. If one of these is missing, CRT is unlikely to benefit the patient. The patterns of the LV deformation strain curves allows a complex assessment of both the timing and contractility [8].

Previous studies have demonstrated that identifying the LBBB-related strain pattern highly predicts the LV reverse remodelling [9]. We believe that a generalized appraisal of dyssynchrony including the analysis of the strain curves from all of the apical views might be a more precise measure to predict the CRT effect. Therefore, we propose a new method, called the Regional Strain Pattern Index (RSPI), which merges information about the dyssynchrony of the strain curves from all three apical views. The objective of this study was to verify RSPI in predicting response to CRT.

## 2. Materials and Methods

### 2.1. Population

The prospective study included 49 subjects (84% male, 66.5 ± 10 years, 34.7%/63.3% in New York Heart Association class II/III) with symptomatic heart failure qualified for CRT implantation in class I/IIa according to the 2013 ESC Guidelines [10]. Firstly, 71 consecutive patients were recruited but twenty-two patients were lost during the follow-up (six patients due to poor echocardiographic window, four patients because of suboptimal pace delivery (BiV < 90%), four patients were lost during the follow-up, four patients declined participate in the study, two had dysfunction of CRT (dislocation of LV lead) and two patients died). The remaining 49 patients constituted the study group. Twenty (40.8%) patients had already had a cardiac implantable electronic device (CIED) and were receiving an upgrade to a resynchronization system, while the others (29 patients, 59.2%) received a CRT de novo. All of the patients had received optimal pharmacological therapy before CRT including renin-angiotensin antagonists (42 patients; 85.7%), β-blockers (48 patients; 98%), loop diuretic (46 patients; 93.9 %), mineralocorticoid receptor antagonist (46 patients, 93.9%). The exclusion criteria included acute coronary syndrome for three months, inadequate CRT delivery after the follow-up (BiV pacing rate < 90%) or a poor image quality of the echocardiography. All study patients had transthoracic echocardiography and NYHA functional class assessment before CRT implantation and after 15 ± 7 months follow-up period. All of the patients were informed and signed a written consent. The study protocol was approved by the local bioethical committee.

### 2.2. Echocardiography—General Data

A full standard echocardiography was performed using a cardiovascular ultrasound system (Vivid 7, GE Medical Systems, Horten, Norway) before and 15 ± 7 months after CRT implantation. LV end-systolic volume (LVESV), end-diastolic volume (LVEDV) and LVEF were measured using the biplane Simpson method. Left Ventricle Global Longitudinal Strain (LVGLS) measurements were performed to assess global left ventricular function. The response to CRT was defined as a ≥15% LVESV reduction after the follow-up period.

The echocardiographic examination included standard gray-scale and color-coded tissue Doppler imaging (TDI) in all of the apical views (4-chamber, 2-chamber and 3-chamber) with six basal and six-midsegmental models of the myocardial velocity curves. Time to peak myocardial velocity during the ejection phase was measured in relation to the beginning of the QRS complex [9].

In addition to the TDI-derived parameters, gray-scale imagining that was optimized for longitudinal strain (with high frame rates; >35 frames/s) and subsequent strain pattern analysis was performed. Two-dimensional longitudinal strain data were processed using a speckle-tracking examination from the apical views and the reference point was the beginning of the QRS complex. The endocardial border was traced in end-systole. Segmentation of the region of interest for a 18-segment model was performed. Segmentation was automatically proposed by the analysis software with visual verification and a manual correction if needed [10].

The ejection phase was defined as the time from the beginning of QRS to aortic valve closure (AVC). AVC was defined using a pulsed-wave Doppler ultrasound in the LV outflow tract from an apical 5-chamber view. The AVC timing was appropriately noted on the tissue velocity and strain waveforms. Three cardiac cycles were taken to average measurements. Echocardiographic images were analyzed offline with a customized software package (EchoPac, General Electric (GE) Healthcare, Boston, MA, USA).

Atrioventricular (defined as the left ventricular diastolic filling time/RR time, LVDFT/RR), interventricular (defined as the difference between the aortic and pulmonary pre-ejection time- interventricular mechanical delay, IVMD) and intraventricular dyssynchrony parameters were evaluated.

The following intraventricular dyssynchrony parameters were calculated: ○Septal flash (0/1)○SPWMD (Septal to posterior wall motion delay, which was assessed using M-mode echocardiography from the parasternal short-axis view at the papillary muscle level) ○Four-chamber max intraventricular delay—maximal difference in the time-to-peak systolic velocity curves among the four sites (two basal, two midventricular) in the 4-chamber apical view○Two-chamber max intraventricular delay—maximal difference in the time-to-peak systolic velocity curves among the four sites (two basal, two midventricular) in the 2-chamber apical view○Three-chamber max intraventricular delay—maximal difference in the time-to-peak systolic velocity curves among the four sites (two basal, two midventricular) in the 3-chamber apical view○Maximum time delay technique—maximal difference in the time-to-peak systolic velocity curves between any two of the 12 LV segments (six basal, six midventricular)○Mechanical dyssynchrony index (Yu index)—standard deviation of the time-to-peak systolic velocity in the 12 LV segments (six basal, six midventricular)○Strain pattern analysis○Regional strain pattern index—RSPI

### 2.3. Echocardiography—Strain Pattern Analysis

Strain pattern analysis was performed according to the model proposed by Risum et al. [9]. The pattern of the strain curve in patients with LBBB (so-called “classical pattern”) reflects an abnormal, dyssynchronous activation of the LV walls. The classical pattern is characterized by an early peak contraction in the early-activated wall, whereas the opposing late-activated wall is prestretched and shows a late contraction [11].

All three criteria must be fulfilled to identify a strain pattern as “classical” (LBBB-related):(1)Early contraction of at least one basal or midventricular segment in septal or anteroseptal wall and early stretching in at least one basal or midventricular segment in the opposing wall,(2)the early peak contraction does not exceed 70% of the ejection phase,(3)the early stretching wall shows a peak contraction after aortic valve closure.

Strain pattern analysis was performed using all three apical views. If all three criteria were fulfilled in at least one view, then the strain pattern was recognized as being “classical”. A pattern that did not fulfill the criteria of a classical pattern was considered to be a “heterogeneous pattern”.

### 2.4. Echocardiography—Regional Strain Pattern Index

Our innovative scoring system is based on the methodology of strain pattern analysis that was proposed by Risum et al. [9]. By definition, the classical pattern includes three criteria, but it comprises four components, actually: (1) the early contraction of the early-activated wall, (2) the prestretching of the opposing, late-activated wall, (3) the contraction of the early-activated wall occurs in the first 70% of the systolic ejection phase and (4) the peak contraction of the late-activated wall occurs after aortic valve closure. Illustration of the four components of RSPI is shown on Figure 1.

The component “contraction of the early-activated wall within the first 70% of the systolic ejection phase” was verified based on the timing measurements (time to peak systolic strain of the early activated wall with regard to the ejection phase time). The remaining three components were based on a visual assessment and were referred to as peak strain curve. The presence of RSPI component was recognized if there was an evident positive/negative peak strain curve. Evaluation of echo strain measurements was performed by one observer. In 20 randomly selected strain studies the inter- and intra-observer variabilities for the longitudinal strain were 7 and 5%, respectively.

RSPI calculation is based on strain curves analysis from apical 4-, 2-, and 3-chamber views. RSPI was calculated as the sum of dyssynchronous components from all three apical views. One point was attributed to the presence of each component, thus a maximum of 12 points could be achieved (four points in each view). Presence of dyssynchronous components was analyzed among basal or midventricular segments. RSPI has no relation to the type of strain pattern (classical or heterogeneous). An example how to calculate RSPI is shown in Figure 2, Figure 3 and Figure 4.

The study population was divided into two groups according to the RSPI score: ≥7 points (19 patients, 38.8% of the population) and <7 points (30 patients, 61.2% of the population). 

### 2.5. Statistical Analysis

The statistical analysis was performed using Statistica 10 Software. The distribution was verified using a Shapiro-Wilk test. Continuous variables were compared using Kolmogorov-Smirnov test or Wilcoxon test when appropriate. The comparison of RSPI score between groups was performed by Mann-Whitney U test. Categorical variables (reported as numbers with percentages) were tested using χ^2^ statistics and the McNemar test when appropriate. Spearman rank coefficients tests were used to determine the relationships between the variables. Univariate regression analysis with dyssynchrony indexes as the independent variables was also performed. The receiver operating characteristics (ROC) was analyzed for RSPI. Reproducibility was assessed for the strain measurements. *p* value < 0.05 was considered to be statistically significant.

## 3. Results

### 3.1. Baseline Characteristics

The study included 49 patients (84% male, 66.5 ± 10 years, NYHA II/III/IV: 34.7%/63.3%/2%; 57.1% ischemic etiology of HF), who underwent CRT implantation. The mean QRS duration was 173.1 ± 19.1 ms. Thirty-five (71.4%) patients had a native LBBB according to the ESC 2013 criteria. Seven (14.3%) patients had a dominant right ventricular pacing rhythm (pacemaker-dependent patients, and therefore native rhythm could not be defined) and seven patients had a non-LBBB morphology. Twenty (40.8%) patients had CIED (three pacemakers (PM) and 17 ICD) and underwent an upgrade to a resynchronization system. Among the three patients with a previously implanted PM, two had a dominant right ventricular pacing rhythm and one had LBBB. Among the 17 patients with ICD, five of them had a dominant right ventricular pacing rhythm, 10 had LBBB and two had non-LBBB. Thirty-two (65.3%) patients had a contraction pattern (classical pattern) that was typical for LBBB and 17 (34.7%) patients had a heterogeneous pattern according to Risum’s method. The baseline characteristics are presented in Table 1. Thirty-six (73.5%) patients responded positively to CRT. No significant differences between responders and non-responders were observed regarding baseline demographics, clinical or echocardiographic parameters. However, non-responders compared with responders were more likely to have CIED before CRT implantation and they underwent “up-grade to CRT” (not de novo implantation).

### 3.2. Follow-Up

The mean follow-up was 14.9 ± 7 months. For the entire study group, a significant decrease in NYHA class was observed (from 2.75 ± 0.52 to 1.9 ± 0.7; *p* < 0.001). There were no important differences in the clinical response between the responders and non-responders (Table 2). During the follow-up, almost all of the echocardiographic parameters improved. The LVEDV and LVESV volumes were reduced. The LVEF increased from 24.9 ± 6.4% to 31.9 ± 6.9%; *p* < 0.001. LVGLS) for the entire population increased from −6.94 ± 2.16 to −7.95 ± 2.68%; *p* = 0.039.

### 3.3. Effect of CRT on the Dyssynchrony Parameters

Table 2 shows the changes in dyssynchrony indexes after CRT. All of the fields of dyssynchrony improved after CRT implantation. The incidence of the septal flash and the typical LBBB contraction pattern decreased (from 31.1% to 11.1%, *p* = 0.039 and from 65.3% to 12.2%, *p* < 0.001, respectively). Similarly, tissue velocity—derived dyssynchrony indexes changed significantly after CRT. A significant reduction of dyssynchrony was observed in the responders but not in the non-responder group.

### 3.4. Regional Strain Pattern Index

The RSPI was higher in the responders than in the non-responders (5.86 ± 2.92 vs. 4.08 ± 2.4; *p* = 0.044). The RSPI was independent of gender (male: 5.15 ± 2.93; female: 6.63 ± 2.34; *p* = 0.21), HF etiology (ischemic vs. non-ischemic: 5.18 ± 2.83 vs. 5.67 ± 2.99, respectively; *p* = 0.72) or the presence of LBBB (LBBB: 5.83 ± 2.68; non-LBBB 4.29 ± 3.15; *p* = 0.15). No correlation was found between the RSPI and the echocardiography parameters, dyssynchrony indexes, QRS duration or NYHA class.

ROC curve analysis was performed for the RSPI (Figure 5). The optimal cut-off value to predict CRT response was at 7 points. RSPI ≥ 7 had a 50% sensitivity, a 92% specificity and a 64.7% positive and 40% negative predictive value in predicting the CRT response (area under curve, AUC = 0.691, *p* = 0.014).

In the RSPI ≥ 7 points group, reverse remodelling (defined as a reduction of LVESV ≥ 15%) was observed in 18/19 patients (94.7%). Among the non-responders, only one patient (1/13; 7.7%) had RSPI of more than seven points and the other 12 patients (92.3%) were in the RSPI < 7 points group (*p* < 0.01). 

### 3.5. Prediction of the Response to CRT

At the 15-month follow-up, the mean LVESV reduction was 23.2 ± 19.8 % (range −22% to 72.3%). None of the baseline dyssynchrony indices correlated with the LVESV reduction. For the entire population, the RSPI did not show a relationship with the reverse remodelling (rho = 0.079; *p* = 0.59). In the univariate logistic regression analysis (Table 3) among all of the dyssynchrony indexes, only the RSPI ≥ 7 points predicted a positive response to CRT (OR: 12; 95% CI = 1.33–108.17; *p* = 0.027).

## 4. Discussion

Numerous approaches have been proposed to quantify dyssynchrony and to improve patient’s selection for CRT implantation [12,13]. In this study, we have presented our innovative proposal for the prediction of CRT response. We have introduced and verified the clinical use of RSPI. Our modified analysis of the contraction strain curve pattern was aimed at minimizing and overcoming the limitations of the previously employed echocardiographic indices of LV dyssynchrony.

RSPI is based on the assumption that dyssynchrony is a quantitative rather than qualitative process and RSPI reflects the severity of the disease. RSPI is expected to be effective also in a scarred myocardium. In the segments with an impaired contractility, the lack of a strain curve makes it impossible to define the strain pattern. However, in such a case, the features of intraventricular dyssynchrony may be present in other segments/views. Theoretically, this remaining portion of dyssynchrony in the viable segments may be revealed by RSPI and is believed to be a potential indicator for a positive response to CRT. Therefore, it is expected that a high RSPI score will identify responders even in case of patients with scarred segments and abnormal regional contractility.

Four important issues are listed below along with an explanation of why RSPI appears to be better than other methods for predicting the beneficial effects of CRT.

Firstly, it is a quantitative approach to dyssynchrony. Some of the classical, aforementioned parameters are based on rigid cut-off values or on the presence of specific signs such as a septal flash. We assumed that dyssynchrony (and the CRT response as well) is a “continuous” value and that it cannot be treated solely in terms of “presence” or “absence”.

Secondly, it is a composite appraisal of dyssynchrony. The advantage of RSPI is that it takes into account dyssynchrony in 12 regions (three apical views with four segments in each view). Previous investigators have shown that the cumulative score of dyssynchrony increases the responsiveness to CRT [14,15]. In the classical parameters, the diminished contractility of one wall may lead to incorrect results. In RSPI, even in the case of impaired myocardial viability in one region, features of dyssynchrony may be present in the remaining regions. Therefore, the final RSPI could remain high, thus reflecting an existing dyssynchrony. Moreover, RSPI reflects changes during the entire cardiac cycle, not just in the ejection period.

Thirdly, the contractility. Patients without preserved contractility are unlikely to have a positive response to CRT. Abnormal contractility modifies the strain-derived pattern of LBBB [16]. Decreased contractility and scarring may predict a poor response to CRT regardless of the pathogenesis of cardiomyopathy [17,18]. Our innovative parameter indirectly assesses the contractility. The absence of contraction makes it impossible to assess the strain curves. Therefore, RSPI would not be scored in these regions (which actually means a “0” point).

Fourthly, the feasibility and simplicity. Many of the proposed dyssynchrony parameters require either specialized software or complex calculations [19]. To calculate RSPI, there is no need for extra tooling or to transfer the images to another system for the final processing of the results. RSPI calculation is possible ad hoc while performing a basic echocardiography examination.

Measurements of various dyssynchrony parameters were performed (Table 3). Among all of the indexes, only RSPI ≥ 7 points showed a significant relation with the CRT response in the univariate model (OR: 12; 95% CI 1.33–108.17; *p* = 0.027), whereas the general RSPI reached a borderline statistical significance (OR: 1.26; 95% CI 0.98–1.61; *p* = 0.068). Therefore, maybe the innovative quantitative methods based on strain analysis would be useful in improving the identification of CRT responders. Recently, Gorcsan et al. [20] proposed Systolic Stretch Index (SSI)—similar to our methodology based on strain analysis. SSI was strongly associated with favorable clinical outcomes including in the important patient subgroup with QRS width 120 to 149 ms or non-LBBB. 

In our study, we intended to prove the usefulness of RSPI in case of all the patients who met the CRT criteria. Our group reflects the real-life, potential CRT recipients. In daily clinical practice, it is not only LBBB but also patients with non-LBBB and patients with CIED that undergo an up-grade procedure. Overall, the positive and negative predictive values of RSPI are quite modest but we must keep in mind that the study population was heterogeneous with different baseline conditions. Within the whole perspective, among the patients with different kinds of CRT indications, RSPI was positively verified with a specificity at satisfactory level of 92.3%.

The arguments presented in the discussion show that the evaluation of dyssynchrony is a composite, multifaceted problem. The Regional Strain Pattern Index is our new approach and has the potential to become a simple and practical tool for everyday use.

## 5. Limitations

Some limitations to our study should be addressed:(1)It was a single center study. The study sample was small and quite heterogeneous (some patients had atrial fibrillation, while some had non-LBBB). Therefore, the predictive value of RSPI should be validated in a larger study with a more homogeneous population.(2)The quality of the echocardiographic examination is crucial for image-based measurements of dyssynchrony. Suboptimal image quality may affect the results. From 71 consecutive patients qualified to enter into the study, finally six (8.5%) patients were excluded due to poor echocardiographic window.(3)Strain pattern methodology relies on visual evaluation of the strain-derived curves. Therefore, during RSPI assessment, in some cases there may be discrepancies between observers.(4)RSPI is a single parameter and reflects intraventricular dyssynchrony, whereas dyssynchrony and the response to CRT are multimodal ones. The selection of the most appropriate candidate for CRT might require a combined approach rather than a single parameter.(5)The QRS duration criterion for entry into the study was 120 ms (according to the previous guidelines), whereas at present the cut-off value is 130 ms. However, no patient had a QRS shorter than 140 ms in our population.(6)Post-implantation CRT optimization was not taken into consideration.

## 6. Conclusions

RSPI constitutes an innovative, valuable predictor for a CRT response. RSPI value of ≥7 points is an independent predictor of CRT response. RSPI appears to be better than the previously used dyssynchrony indexes for selecting patients for CRT and for efficiently identifying long-term responders. 

## Figures and Tables

**Figure 1 ijerph-18-00926-f001:**
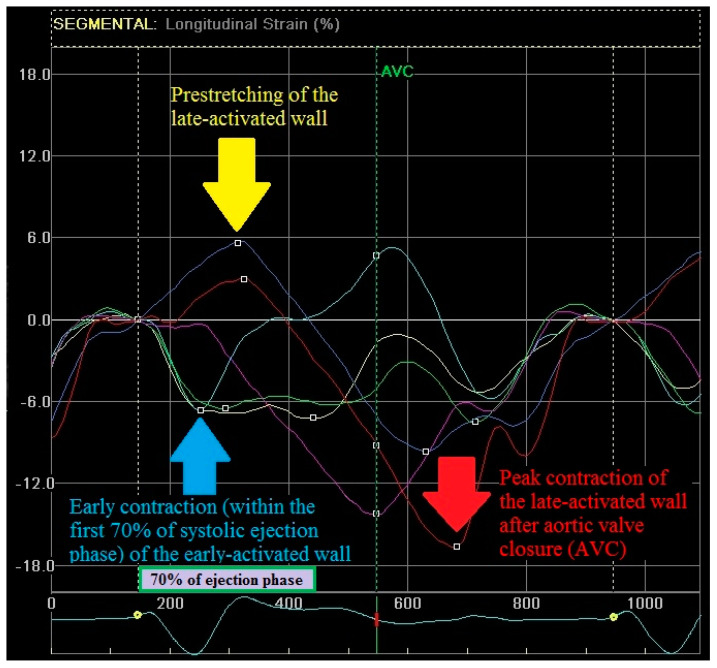
Components of Regional Strain Pattern Index (RSPI). Illustration of the four components of RSPI based on an analysis of the 4-chamber apical view strain curves: (1) the early contraction of the early-activated wall (light blue line; light blue arrow); (2) the prestretching of the opposing, late-activated wall (blue line, yellow arrow); (3) the contraction of the early-activated wall occurs in the first 70% of the systolic ejection phase and (4) the peak contraction of the late-activated wall occurs after aortic valve closure (red line, red arrow). AVC indicates aortic valve closure.

**Figure 2 ijerph-18-00926-f002:**
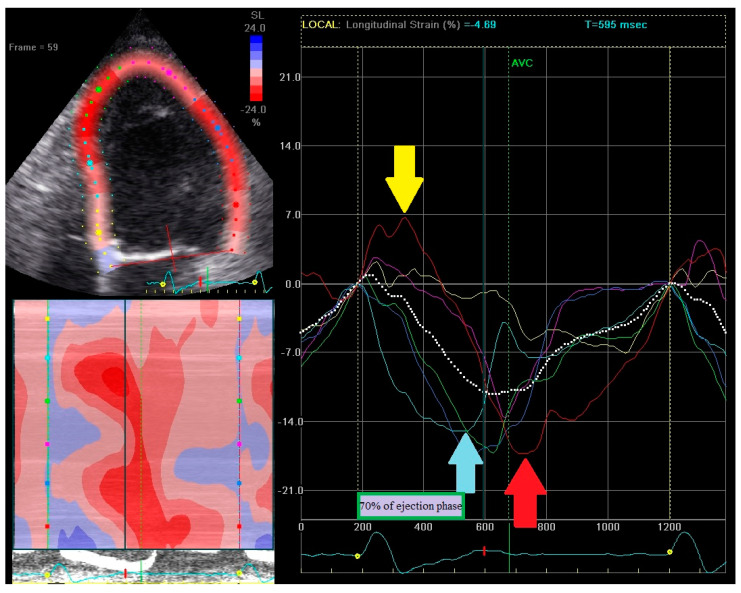
RSPI calculation: 4-chamber view. In 4-chamber view patient reached 3 points: early contraction of the midventricular segment (light blue line; light blue arrow) in the interventricular septum [1 point] but peak contraction slightly exceeds 70% of the systolic ejection phase [0 point]; early prestretching of the basal segment (red line; yellow arrow) in the lateral wall [1 point] with the peak contraction (red arrow) after aortic valve closure [1 point].

**Figure 3 ijerph-18-00926-f003:**
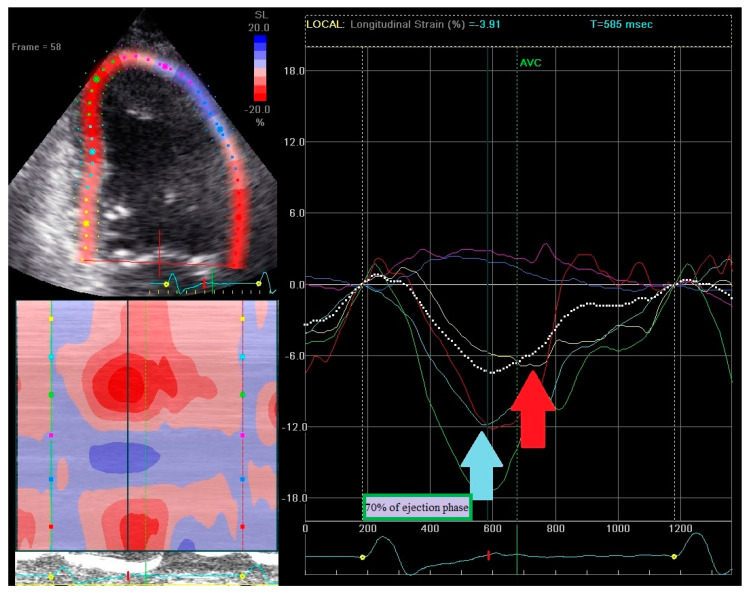
RSPI calculation: 2-chamber view. In 2-chamber view patient reached 2 point: the basal segment in the anterior wall exhibits contraction movement (red line; light blue arrow) [1 point], but it does not fulfill criterion of early 70% of the ejection phase [0 point]. The basal segment in the inferior wall (yellow line) does not show early stretching [0 point] but peak contraction occurs after aortic valve closure (red arrow) [1 point].

**Figure 4 ijerph-18-00926-f004:**
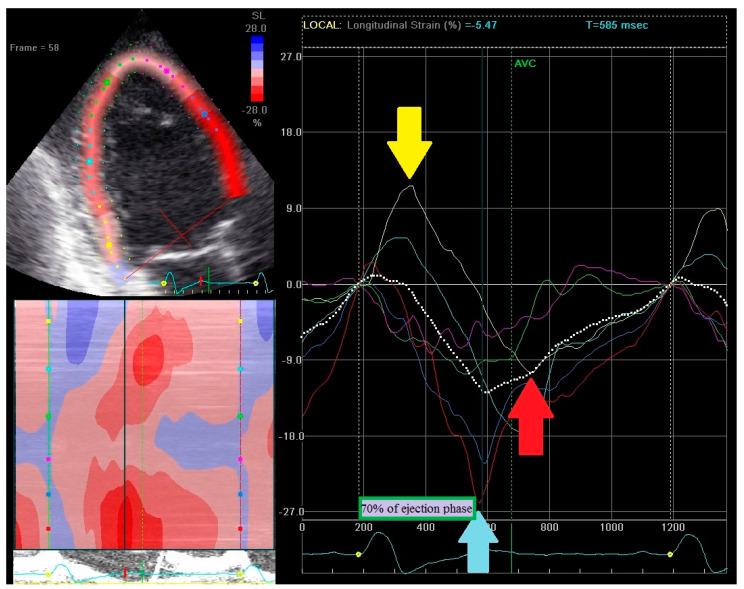
RSPI calculation: 3-chamber view. In 3-chamber view patient reached 3 points: early contraction of the basal segment (red line; light blue arrow) in the anteroseptal wall [1 point] but peak contraction exceeds 70% of the systolic ejection phase [0 point]; early prestretching of the basal segment (yellow line; yellow arrow) in the posterior wall [1 point] with the peak contraction (red arrow) after aortic valve closure [1 point].

**Figure 5 ijerph-18-00926-f005:**
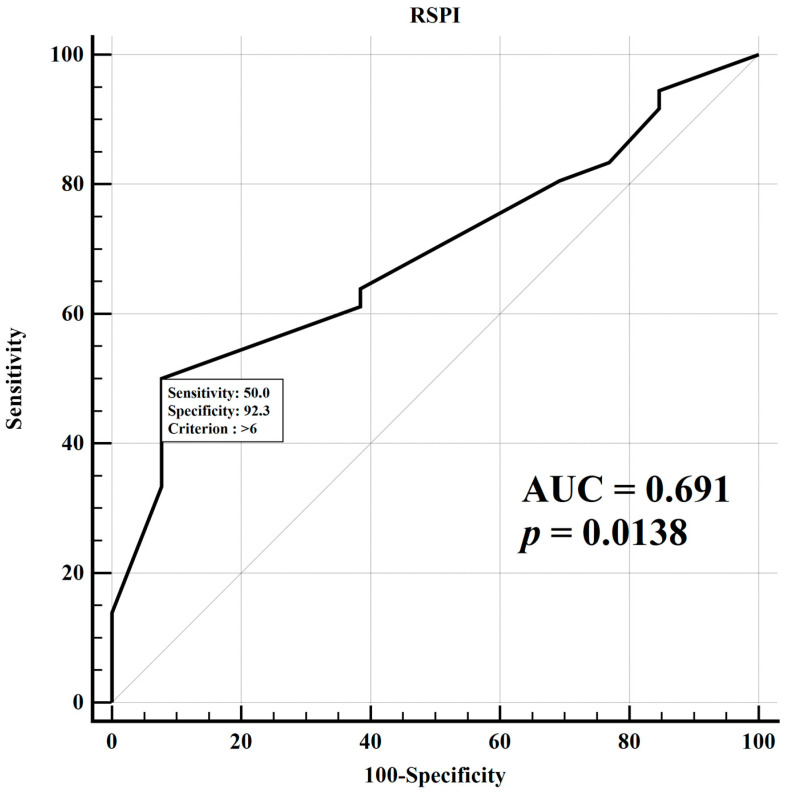
Receiver operating characteristic curves for Regional Strain Pattern Index (RSPI) in prediction of Cardiac Resynchronization Therapy (CRT) response (AUC = area under the curve).

**Table 1 ijerph-18-00926-t001:** Baseline characteristics of general population, responders and non-responders.

	Study Population (*n* = 49)	Responders (*n* = 36)	Non-Responders (*n* = 13)
Age (years)	67 ± 10	68 ± 10	63 ± 10
Male Sex, *n* (%)	41 (84)	30 (83.3)	11 (84.6)
NYHA Functional Class	2.8 ± 0.5	2.8 ± 0.6	2.7 ± 0.4
Baseline NYHA Class III, *n* (%)	31 (63.3)	21 (58.3)	10 (76.9)
Ischemic Etiology of HF, *n* (%)	28 (57.1)	20 (55.6)	8 (61.5)
QRS (ms)	173 ± 19	173 ± 21	174 ± 16
LBBB, *n* (%)	35 (71.4)	27 (75)	8 (61.5)
AF at Implantation, *n* (%)	7 (14.3)	6 (16.7)	1 (7.7)
CIED Before CRT (= up-grade to CRT), *n* (%) *	20 (40.8)	11 (30.6)	9 (69.2)
LVESV (mL)	218 ± 109	217 ± 107	223 ± 119
LVEF (%)	25 ± 6	24 ± 6	27 ± 7

Note: NYHA: New York Heart Association; HF: Heart Failure; QRS = QRS width; LBBB—Left Bundle Branch Block; AF: Atrial Fibrillation; CIED: Cardiac Implantable Electronic Device; CRT: Cardiac Resynchronization Therapy; LVESV: Left Ventricular Ventricular End Systolic Volume; LVEF: Left Ventricular Ejection Fraction; * *p* < 0.05 responders vs non-responders.

**Table 2 ijerph-18-00926-t002:** Dyssynchrony indexes and NYHA class before and after CRT in responders and non-responders.

	Responders			Non-Responders		
	Baseline	After CRT	*p* Value	Baseline	After CRT	*p* Value
Echocardiographic Parameters						
LVEF (%)	24 ± 6	34 ± 7	<0.001	27 ± 7	27 ± 5	0.433
LVESV (mL)	217 ± 107	147 ± 87	<0.001	223 ± 119	218 ± 110	0.6
LVEDV (mL)	277 ± 127	218 ± 107	<0.001	298 ± 139	294 ± 142	0.35
Dyssynchrony Indexes						
LVDFT/RR (%)	40.5 ± 9	49.3 ± 6.6	<0.001	43.5 ± 9.5	43.8 ± 8.2	0.753
Interventricular Mechanical Delay (IVMD) (ms)	36.7 ± 36	13.2 ± 19.5	<0.001	36.9 ± 31.7	14.6 ± 22	0.025
Septal Flash, *n* (%)	11 (32.4)	2 (5.9)	0.016	3 (27.3)	3 (27.3)	0.617
SPWMD (ms)	82.3 ± 176.5	−44.3 ± 101.2	0.002	85.4 ± 116	20 ± 194.4	0.424
4-chamber Max Intraventricular Delay (ms)	99.2 ± 80.5	100.6 ± 84.8	0.812	123.8 ± 101.5	108.3 ± 137.4	0.456
Maximum Time Delay Technique (ms)	168.3 ± 107.7	137.8 ± 74.8	0.164	204.6 ± 93	162.5 ± 113.7	0.196
Maximal Opposing Wall Delay (ms)	144.7 ± 94.3	123.3 ± 76.5	0.21	177.7 ± 92	139.2 ± 117.2	0.21
Yu Index (ms)	60.1 ± 40.6	46.9 ± 24.3	0.109	71.6 ± 34.4	53.8 ± 31.2	0.158
Classical Pattern (0/1), *n* (%)	11/25 (30.6/69.4)	32/4 (88.9/11.1)	<0.001 (McNemar’s test)	6/7 (46.2/53.8)	11/2 (84.6/15.4)	0.13 (McNemar’s test)
RSPI	5.86 ± 2.9	2.69 ± 2.3	<0.001	4.08 ± 2.4	2.31 ± 2.2	0.083
Clinical Response						
NYHA Class	2.8 ± 0.6	1.9 ± 0.7	<0.0001	2.7 ± 0.4	2 ± 0.7	0.005

Note: LVEF: Left Ventricular Ejection Fraction; LVESV: Left Ventricular Ventricular End Systolic Volume; LVEDV: Left Ventricular Ventricular End Diastolic Volume; LVDFT/RR: Left Ventricular Diastolic Filling Time/RR ratio: SPWMD Septal-To-Posterior Wall Motion Delay; RSPI: Regional Strain Pattern Index; NYHA: New York Heart Association; CRT: Cardiac Resynchronization Therapy.

**Table 3 ijerph-18-00926-t003:** Dyssynchrony indexes: univariate logistic regression analysis for CRT response.

	Univariate Logistic Regression Analysis (Responder: ∆LVESV ≥ 15%)
	Odds Ratio [OR]; 95% Confidence Interval. (‘*p*’ Value)
**Atrioventricular Dyssynchrony**	
LVDFT/RR (Left Ventricular Diastolic Filling Time/RR time) (%)	0.963; 95% CI = 0.89–1.038 (*p* = 0.32)
LVDFT/RR < 40% (0/1)	0.5; 95% CI = 0.12–1.98 (*p* = 0.31)
**Interventricular Dyssynchrony**	
IVMD (Interventricular Mechanical Delay) (ms)	0.999; 95% CI = 0.98–1.02 (*p* = 0.99)
IVMD ≥ 40 ms (0/1)	1.07; 95% CI = 0.29–3.96 (*p* = 0.92)
**Intraventricular Dyssynchrony**	
Septal flash (0/1)	1.56; 95% CI = 0.34–7.17 (*p* = 0.56)
SPWMD (Septal to Posterior Wall Motion Delay) (ms)	0.999; 95% CI = 0.995–1.004 (*p* = 0.95)
4-chamber max intraventricular delay (ms)	0.99; 95% CI = 0.989–1.004 (*p* = 0.39)
Maximum Time Delay Technique (ms)	0.997; 95% CI = 0.99–1.003 (*p* = 0.29)
Maximal Opposing Wall Delay (ms)	0.996; 95% CI = 0.99–1.003 (*p* = 0.29)
Yu Index (ms)	0.993; 95% CI = 0.977–1.009 (*p* = 0.37)
**Strain Pattern Analysis and RSPI**	
General Classical Pattern (0/1)	1.95; 95% CI = 0.51–7.4 (*p* = 0.32)
RSPI (General Score)	1.26; 95% CI = 0.98–1.61 (*p* = 0.068)
RSPI ≥ 4 points	1.84; 95% CI = 0.42–8.06 (*p* = 0.41)
RSPI ≥ 7 points	12; 95% CI = 1.33–108.17 (*p* = 0.027)

Note: RSPI: Regional Strain Pattern Index.

## Data Availability

Not applicable.

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
