# Peer review of "Regional Strain Pattern Index—A Novel Technique to Predict CRT Response"

_ijerph, 2021, doi:10.3390/ijerph18030926_

Round 1

Reviewer 1 Report

The study addresses the never ending story of CRT prediction. In this trial Authors used a novel index Regional Strain Pattern Index (RSPI). They showed that responders reached higher mean RSPI values than non-responders. In logistic regression analysis a value of RSPI ≥7 points was a predictor of favorable CRT. There are a few issues that need to be addressed:

  1. There are no data on the feasibility of the method. Authors state clearly that suboptimal imaging might be a problem but they do not tell us what is the feasibility in their expert hands.
  2. They have a loss to follow-up >30%. This is too much. Please address and explain why. Moreover this clearly introduces a selection bias.
  3. Authors state that the prediction CRT should be based on quantitative and not qualitative parameters. However, they use a cut off of RSPI set at 7. It would be interesting to check the model by using individual indexes as with NYHA class.
  4. There are no data on outcome. Since they followed-up patients it would be interesting to know the event rate in these patients. These data would strengthen the manuscript.
  5. The discussion should be more focused on the novelty of the method and the clinical impact of these results.

Reviewer 2 Report

In this article, authors have purposed regional strain pattern index (RSPI) as a novel technique to predict cardiac resynchronization therapy response (RSPI ≥7 points, 12, p=0.027). RSPI could be beneficial for heart failure patients as a novel predictor for CRT response. This paper falls under the scope of journal as research is done related to public health aspect. I recommend author to address few issues before considering it appropriate for publications.  

Comments

  1. Section 1. Introduction, line 31: please specify what are those “symptoms”
  2. Section 1. Introduction, Line 32, what is QRS? Line 34 LV?, Line 41 LBBB? Please mention full name if it appears for the first time in manuscript.
  3. Introduction first sentences “Cardiac resynchronization therapy (CRT) improves the outcome and reduces symptoms in advanced Heart Failure with a Reduced Ejection Fraction (HFrEF) patients with a prolonged QRS”. It would be good if authors add information to support this sentence with some statistic showing data/number/percentage of HFrEF patient improved with CRT.
  4. Introduction is too short, needs more elaboration highlighting why RSPI is superior to other predicting tolls in CRT response. Authors can also include the limitation of other (if any) similar prediction tools.
  5. Figure 1. Please change the colour of text as it is not clearly visible (specially with blue and red)
  6. Texts are cryptic in many places. Needs extensively correction for structure, spelling, meaningful/clarity of sentences.
  7. Please remove CRT from keywords. Cardiac resynchronization therapy itself is CRT. Abbreviation is generally excluded as keywords.

Round 2

Reviewer 1 Report

Authors have addressed all the issues raised 

Author Response

I would like to thank you for your review, which helped to improve the manuscript.

I remain at your disposal.

Best regards

Michal Orszulak

Reviewer 2 Report

Thank you for addressing all of my comments. The revised manuscript looks better than previous version. I have recommended for publication 

Author Response

Dear Reviewr,

I would like to thank you for your review, which helped improve the manuscript. I would like also to express my gratitude for recommendation for publication.

I remain at your disposal

Best regards

Michal Orszulak